# Validating an Assessment Tool for Oral Health and Oral Care Procedures Performed by Healthcare Workers for Older Residents in Long-Term Care Institutions

**DOI:** 10.3390/healthcare12050558

**Published:** 2024-02-28

**Authors:** Florence M. F. Wong, Anna Wong, Wai Keung Leung

**Affiliations:** 1School of Nursing, Tung Wah College, Hong Kong SAR, China; annawong@twc.edu.hk; 2Division of Periodontology and Implant Dentistry, Faculty of Dentistry, The University of Hong Kong, Hong Kong SAR, China; ewkleung@hku.hk

**Keywords:** aged, healthcare workers, long-term care, oral care, oral health

## Abstract

Poor oral health is a growing concern among older populations. It is often caused by a failure to maintain proper oral hygiene and inaccessible dental care. Poor oral health in older individuals in long-term care institutions (LTCIs) can be attributed to the fact that healthcare workers might be poorly trained in oral care assessment and practice. To address this issue, an assessment tool has been developed and validated to guide and evaluate healthcare workers’ oral care practices, ensuring the delivery of adequate care and early detection of dental diseases in LTCIs. The tool includes an oral health assessment and an assessment of oral care procedures. It was developed following a robust literature review, two stages of expert reviews, content validity checks, and a pilot study. A total of twenty-three items were developed and validated, with seven items related to oral health assessment and sixteen related to oral care procedures. The items were assessed for content validity and relevance, with high values of 1 obtained for all Item-level Content Validity Index (I-CVI), Scale-level Content Validity Index (S-CVI), and S-CVI/Universal Agreement (UA) scores. This indicates a high level of agreement among the experts (*n* = 12) regarding the relevance and importance of the items. A pilot study involving 20 nursing students confirmed the tool’s reliability, applicability, and feasibility, demonstrating its high appropriateness and applicability. The newly developed and validated assessment tool can effectively guide and evaluate healthcare workers’ oral care practices, enhancing their competence and improving the oral health of older residents.

## 1. Introduction

### 1.1. Background

The global ageing population is a growing concern, with an estimated 1.4 billion people aged 60 and older by 2030, doubling to 2.1 billion by 2050 [1]. In Hong Kong, the number of older individuals is expected to reach 2.74 million by 2046 [2]. Poor oral health among this population has become a challenge as it is often overlooked, negatively impacting their general health and quality of life [3,4]. Individuals aged 65 years and older experience a range of mild to severe oral health problems, including untreated dental caries, gum diseases, complete tooth loss, and even oral cancer [5,6,7]. In Hong Kong, about 5.6% of the population aged 65–74 living in the community are completely edentulous, which has been attributed to the lower cost associated with tooth extraction than other options [7]. Over 75% of older individuals in the community and nearly all residents in long-term care institutions (LTCIs) do not attend annual dental check-ups unless they have oral health problems [3,7].

The poor oral health of older individuals is closely associated with deteriorating general health conditions and increased morbidity and mortality rates [8]. Poor oral health is linked to non-communicable diseases such as diabetes, heart diseases, chronic obstructive pulmonary disease, nutritional problems, arthritis, and degenerative neural diseases [6,7,8,9,10]. According to the 2017 Global Burden of Disease study, oral disorders could be found related to disabilities in populations aged 50–74 years worldwide [11]. Most people aged 65 and older in developing countries, such as Indonesia and Honduras, had a higher prevalence of dental problems [12]. A study conducted in Hong Kong revealed that individuals aged 65–74 experience multiple oral problems, including dental caries, root caries, gum bleeding, and periodontitis [13]. These issues can result in functional difficulties such as eating and chewing problems, speech impairments, and nutritional deficiencies, ultimately impacting oral health-related quality of life and general health [9,10,14]. Poor oral health in the older population has to be given more attention as this global health concern is more likely to be found in those who belong to low socioeconomic groups, lack insurance coverage, and are ethnic minorities, amplifying the challenges of treating and preventing oral problems using the dental and general healthcare services systems [1,3,15].

To promote oral health in older populations, policies have been developed to provide oral health services support for older individuals in communities and LTCIs in developing and developed countries. For example, in Australia, older people with low income can receive support for public dental care, and oral health therapists are assigned to LTCIs [16]. In some countries, such as Japan, the medical insurance system is adapted to provide oral health services for older adults. In the United Kingdom, the National Institute for Health and Care Excellence (NICE) guideline advocates the maintenance and improvement of the oral health of care home residents [17]. In Hong Kong, various oral and dental health promotion schemes have been developed to address the health quality of the older population, including healthcare vouchers for multiple dental consultations, services, and the Community Care Fund Elderly Dental Assistance Program [18]. However, older individuals residing in LTCIs may face physical constraints and are particularly vulnerable to poor oral health [5,6,19], which is considerably worse than those who live in the community [5,13,20]. This can be attributed to inadequate self-care or healthcare worker-assisted oral care practices [21,22,23,24,25,26].

The deteriorating oral health of older individuals in LTCIs is often associated with insufficient knowledge, attitudes, and practices (KAP) of healthcare workers [27]. Although educational programmes have been implemented to enhance healthcare workers’ oral care knowledge and skills, their effectiveness in oral healthcare delivery exhibited inconsistent results. A case-control study was conducted to investigate the impact of an educational programme on oral health status and care delivery by healthcare workers in LTCIs [28]. The programme aimed to improve the KAP of healthcare workers responsible for oral care in LTCIs. The intervention group exhibited improved knowledge, while the control group demonstrated a significant decrease in knowledge. The oral conditions of the older residents were noted to be poor during oral examinations after oral care procedures performed by healthcare workers who had received oral care education. Despite their partial independence, the older residents were found to have multiple oral problems, such as poor oral hygiene, tooth fractures, periodontitis, draining sinuses, food debris/impaction, and bad breath were observed. Insufficient oral health assessments and poor oral care techniques were identified as contributing factors to the poor oral health of the older residents.

Oral health assessments are crucial for identifying problems early and preventing more severe oral and general health conditions [29]. The widely recognised Oral Health Assessment Tool (OHAT) assesses oral health conditions across eight categories for screening purposes: soft tissue conditions of the lips, tongue, or gums; saliva; natural teeth; dentures; oral hygiene; and dental pain. However, it requires adequately trained nurses or medical doctors to perform it [30]. Healthcare workers in LTCIs may lack the training to conduct the OHAT. Therefore, a simplified and user-friendly health assessment tool appears more appropriate for these healthcare workers [29]. While various oral care checklists exist, no validated assessment tool has been specifically designed for healthcare workers in LTCIs. A validated assessment tool for healthcare workers is necessary to ensure a comprehensive assessment of oral health and oral care procedures among older residents of LTCIs.

This study aimed to develop and validate an assessment tool that can improve current oral care practices and ultimately enhance the oral health of older residents. Once validated, this tool could be used as a routine assessment tool for older residents to be applied by LTCI healthcare workers involved in oral care practices. By having a validated assessment tool that provides guidance on proper oral health assessment and care, healthcare workers could ensure a more comprehensive evaluation of oral health and oral care procedures for older individuals in LTCIs, leading to improved oral and overall health outcomes. 

### 1.2. Operational Definitions

Oral health assessment refers to evaluating a patient’s oral health status, which involves assessing the condition of their teeth, gums, tongue, and other oral structures and appliances [29].Oral care practice refers to the activities involved in maintaining good oral hygiene and health, such as brushing, flossing, interdental brushing, using mouthwash, and facilitating regular dental visits [31].Oral care performance refers to an individual’s ability to carry out oral care procedures effectively and efficiently. This may involve using the proper techniques and tools to clean the teeth, gums, and appliances and identifying and addressing any issues or concerns that arise during the process. This, in turn, helps maintain optimal oral health and prevents dental problems [31,32].

### 1.3. Study Aim

This study aimed to develop and validate an assessment tool to improve current oral care practices delivered by healthcare workers to enhance the oral health of older LTCI residents.

## 2. Materials and Methods

### 2.1. Study Design

This study assessed the content validity of the tool following a five-step tool validation procedure: (1) development of a preliminary assessment tool in the native language based on a literature review, (2) translation and back-translation of the assessment tool, (3) expert consultations, (4) validation of the tool using a pilot study, and (5) finalising the assessment tool [33,34,35].

### 2.2. Literature Review

The preliminary assessment tool in English was developed based on a comprehensive literature review and expert consultation. The aim was to identify potential sources that may inhibit proper oral care procedures and ensure that these procedures effectively prevented errors or mistakes. To develop the preliminary assessment tool, a literature search was conducted to identify studies that met the inclusion criteria: (1) studies published between 2013 and 2023, (2) primary studies that evaluate oral health and/or oral care practices, (3) studies involving older people or LTCI residents, (4) studies with available abstracts, and (5) studies written in English or Chinese that were available on electronic databases. Clinical guidelines or recommendations, editorials, and expert opinion reports were excluded. 

The comprehensive search involved several electronic databases, including PubMed, MEDLINE (OvidSP), and CINAHL. Relevant keywords were identified in the titles, abstracts, or subject descriptor/MeSH terms used. Chinese studies using Chinese keywords were also searched. A search of Google Scholar was carried out, and the reference lists of the relevant studies were hand-searched for other potentially relevant studies based on their titles. The keywords used included “oral health”, “oral care”, “assessment”, “checklist”, “old”, “person”, “people”, “resident”, “community”, “residential home”, and “long-term care institution”. Two reviewers (FMFW and AW) independently screened the list of potential studies and excluded irrelevant publications. In cases of disagreement, a third reviewer (WKL) individually reviewed the papers and applied the above criteria to decide whether a paper should be excluded or retained.

### 2.3. Two-Stage Expert Consultation 

Two expert consultations were conducted. In the first consultation, at least five experts were invited to provide feedback independently on the assessment tool draft (both the English and Chinese versions). They evaluated the tool’s overall format, rating method, and items and provided comments to ensure the accuracy, relevance, and completeness of the oral health assessment and oral care procedures items [36]. The draft was modified based on their comments.

In the second expert consultation, another 5 or more experts were invited to participate in a formal discussion to provide feedback and finalise the assessment tool (English or Chinese version).

### 2.4. Tool Translation and Interviews

Based on the findings of the literature review and the first expert consultation, the appropriate assessment items for oral health and oral care practices were identified. Subsequently, a preliminary assessment tool was developed in both English and Chinese. The tool underwent a translation process to ensure accuracy, including back-translation until both versions were approved by two language reviewers. The language reviewers included a native English speaker and a native Chinese speaker, who were proficient in both languages [37].

### 2.5. Content Validity

During the second expert consultation, the panel of experts assessed the content validity of the assessment tool. They evaluated the appropriateness, structure, clarity, and ambiguity of each item in the preliminary version using the Content Validity Index (CVI) and Content Validity Ratio (CVR). The CVR is calculated by evaluating experts’ ratings regarding the relevance or essentiality of each item in relation to the measured construct. The CVI assesses the extent to which the items in the tool are relevant and clear for measuring the assessed construct. It provides an overall measure of content validity at the item level [38,39,40]. 

To determine the CVI for the relevancy and clarification of each item, the experts rated the items on a four-point scale (1 = not relevant, 2 = somewhat relevant, 3 = quite relevant, 4 = highly relevant). Items with a rating lower than 3 were rephrased based on the experts’ comments. There are two CVI scores used for assessment tool development: the item-level CVI (I-CVI) and scale-level CVI (S-CVI) scores [38,39]. The I-CVI is the proportion of experts who rate an item as ‘relevant and clear,’ with values ranging from 0 to 1. I-CVI values greater than 0.79 indicate a relevant item, values between 0.7 and 0.79 indicate that an item needs revision, and values below 0.7 indicate an item should be eliminated [38]. The S-CVI can be calculated using the Average Content Validity Index (A-CVI) or the Universal Agreement Method (UA). S-CVI values range from 0 to 1, with higher values indicating more significant agreement among experts regarding the relevance and clarity of the items in the tool. A S-CVI/UA of 0.8 or greater indicates excellent content validity [39,40].

Another analysis used to measure the essentiality of an item is the content validity ratio (CVR) [37,39]. The CVR formula is CVR = (Ne − N/2)/(N/2), where Ne represents the number of experts indicating an item to be ‘essential’, and N is the total number of experts [39]. 

### 2.6. The Pilot Study

A concurrent reliability test was conducted to evaluate the level of agreement between two assessors regarding the tool. To test the practicality and feasibility of the assessment tool in real-life settings, its face validity was evaluated using a pilot study involving two experienced assessors, registered nurses, and nurse educators. One assessor had extensive nursing experience and was responsible for clinical placement arrangements and student performance in clinical teaching. The other was an associate professor specialised in skill teaching, particularly for oral health assessment and oral care procedures, and who had conducted research focused on the oral health of older residents in LTCIs. Both assessors used the same assessment tool to evaluate the same oral care procedures, compare their results, and determine their agreement level. The aim was to ensure that all essential and accurate aspects were covered, the items were appropriately worded and organised logically, and they were easily understood and acceptable. The pilot study helped identify potential issues, refine procedures, and gather and act upon feedback before implementing the new tool with the participating subjects [41,42].

For participant recruitment, approximately 15–30 nursing students who met the following criteria were selected for participation in the pilot study: (1) aged 18 or older, (2) were either nurse students who had completed oral care training or licensed healthcare workers responsible for oral care procedures at their workplaces and (3) could communicate in either Chinese or English. The recruitment process used purposive and convenience sampling methods. Participants were asked to perform oral health and care procedures at a tertiary healthcare training institution in a laboratory setting. The assessment tool was employed to document the oral health and care procedures after the subjects were asked for feedback on enhancing and clarifying the assessment tool.

#### Pilot Study Procedure

After obtaining approval from the research ethics committee of the tertiary healthcare training institution, eligible participants were recruited using purposive and convenience sampling methods. All subjects were required to sign an informed consent form before participating in the study. They were instructed to pair up, and a 30 min session was scheduled for each procedure. The paired subjects were expected to perform oral health assessments and oral care procedures on each other. The principal investigator (PI) (FMFW) and co-investigator (Co-I) (AW) served as assessors, utilising the same assessment tool to evaluate the performance of the procedures independently. Their results were compared to determine the level of agreement between them and to assess the appropriateness and adequacy of the assessment items. This approach ensured the accuracy and consistency of the assessments used. Additionally, the PI sought feedback from the participants upon procedure completion to identify areas for improvement. The data collection stopped once the assessment tool was finalised, and no further modifications were deemed necessary. The finalised assessment tool underwent review by the experts involved in the second expert consultation.

### 2.7. Assessment Tool Finalisation

Upon completing the pilot-testing phase, incorporating the feedback, and making the necessary revisions, the assessment tool underwent a comprehensive refinement process. This process ensured that all items were accurate, appropriate, and understandable. The finalised assessment tool was deemed suitable for implementation, contributing to assessing oral health and oral care procedures.

### 2.8. Ethical Considerations

Ethical approval was obtained from the research ethics committee of the study institution (REC2023179) before the study commenced. All eligible participants provided informed consent and were assured that their personal data would be handled with strict confidentiality. Data were encrypted and stored on the PI’s computer. Only the research team had access to the data.

## 3. Results

### 3.1. Content, Domain Specification, and Item Generation

A comprehensive literature review was conducted by searching the MEDLINE, PubMed, and CINAHL databases. In total, 156 articles were initially identified. After removing 72 duplicates, 84 articles remained. Hand-searching was also conducted; however, no additional articles were identified. The titles and abstracts of the 84 articles were screened to ensure their relevancy and adequacy in developing the draft of the assessment tool for oral health and oral care procedures. Based on this screening process, 22 articles were selected for further eligibility review. Two independent reviewers carefully reviewed these 22 articles based on the inclusion criteria and engaged in discussion about their inclusions. Eight articles, all of which were quantitative studies, were selected for further review of their content, domain specification, and item generation. The search procedure and relevant articles selected are illustrated in Figure 1.

The assessment tool consists of two parts in both English and Chinese. The first part (Part I) was developed for oral health screening for older residents in LTCIs with reference to relevant articles. Several oral health assessment tools were identified during the literature review, including the Oral Health Assessment Tool (OHAT), the Kayser-Jones Brief Oral Health Status Assessment (BOHSE), and several modified or newly designed tools. The OHAT was instrumental in determining oral health conditions in the included studies. The assessment items in the BOHSE were similar to those in the OHAT. Therefore, the oral health assessment tool was developed by WKL and FMFW based on OHAT and BOHSE. The appropriateness of the items from the articles, essential items for oral health screening, and the applicability of the items for LTCI healthcare workers conducting regular oral health assessments in older residents were considered. The assessment tool’s first draft of Part I on oral health consisted of various assessment parts, such as lips, oral mucosa, teeth, tongue, and oral-related conditions. If any abnormal oral conditions were observed during the assessment, such as pain, colour, dryness, and swelling, it was recommended that they be reported to the in-charge nurse and a dentist consultation be arranged. The second part (Part II) was designed to assess oral care procedures to ensure proper oral care was used to maintain older residents’ oral health. This section included essential items for each step of the oral care procedure, serving as a reminder or checklist to guide healthcare workers in performing the procedure systematically. Therefore, it can be used as a self-evaluation tool for healthcare workers to review their oral care skills and as an oral care skill evaluation checklist for healthcare trainees, and any unsatisfactory or incomplete items should be reviewed. Guidance and feedback should be provided for their improvement. This section complements Part I, serving as a reminder to healthcare workers to perform oral health assessments to identify potential oral problems in older residents early. 

### 3.2. Two-Stage Expert Review 

Two expert reviews were conducted to ensure the assessment tool’s applicability and appropriateness. In the first review, English and Chinese drafts of the assessment tool for oral health and oral care procedures were reviewed by six experts, including two dentists, one geriatric nurse, one nurse educator, and two dental hygienists. The experts provided comments and suggestions via email or telephone. Modifications were made based on their feedback to align the assessment tool with essential criteria for assessing the oral health conditions of older residents and current practices for oral care procedures. For instance, Item 2, ‘Use standard precautions and appropriate infection control measures during oral care’, was moved to the former part of the checklist to remind healthcare workers about infection control regulations during the procedure. Similarly, Item 5, ‘Ensure the oral cavity is visible under appropriate lighting for oral care and assessment’, was crucial for reminding healthcare workers to use sufficient lighting during the procedure. Item 10, ‘Ensure safety during oral care to all parties involved’, emphasised the importance of following safety precautions to protect the healthcare worker and older resident during the procedure.

In the second review, a panel of another six experts, including a community dentist, a geriatric nurse, a nurse educator, two dental nurses, and a nurse from a residential home, evaluated the tool independently. They assessed the appropriateness, structure, clarity, and ambiguity of each item using the preliminary version of the CVI. The CVR scores were calculated by evaluating the experts’ ratings regarding the relevance or essentiality of each item. Using a four-point rating scale, the experts reviewed and rated all twenty-three items, including seven related to oral health assessment and sixteen related to oral care procedures. Their scores ranged from 3 to 4, indicating that all items were highly relevant. Based on their narrative comments, modifications were made to the items to improve their clarity and precision. Some modifications were related to word use, such as changing ‘whiteness’ to ‘pale’ to more accurately indicate the lip and oral mucosa colour. Additionally, the nurse from the residential home suggested adding items related to oral hygiene conditions, such as food debris trapped between teeth and bad breath, to enhance the appropriateness and practicability of the oral health assessment section (Part I).

The six experts also independently rated each item on a scale of relevance. As all experts rated all items as ‘very relevant’, the I-CVI score for each item was 1.0, indicating excellent agreement among the experts on the relevance and clarity of each item. The CVR was also 1, indicating that all experts consider all items essential. The S-CVI/UA was calculated as the proportion of items all experts rated as relevant and clear. All experts rated all items as ‘very relevant’, indicating excellent agreement among the experts regarding the relevance and clarity of all items in the tool. 

The developed assessment tool demonstrated excellent content validity. Appendix A illustrates the expert review results, while Appendix A shows the CVR and I-CVI results.

### 3.3. Tool Refinement

The Delphi method was used to conduct two rounds of evaluations [39]. In the first round, items were rephrased to improve their clarity, moved to a more suitable place in the assessment to fit the sequence of the oral care procedure, or deleted if they were duplicated or deemed not essential. Five items in Part I and eight items in Part II were rephrased. One item was added to Part I, and three items were added in Part II. Two items in Part II were moved to more suitable places in the oral care procedures. After the CVI scores were calculated and cognitive interviews conducted, 21 items were evaluated and included in the final version of the tool (Appendix A). 

### 3.4. The Pilot Study

A pilot study was conducted to examine the applicability and feasibility of the tool, during which the participants were asked if they encountered any difficulties in understanding or answering the items. Twenty subjects, who were nursing students or enrolled nurses with oral care training, were recruited for the pilot study. Of these, eight were female (40%). The mean age was 24.2 (standard deviation [SD] 3.74). All subjects underwent at least one year of oral assessment and care procedure training. The participants were enrolled in two nursing programmes: a higher diploma in nursing programme for enrolled nurse licensure (*n* = 10) or a bachelor’s degree for a healthcare science programme for registered nurse licensure (*n* = 10, eight were practising enrolled nurses). All were in year two of their programmes. 

All participants were required to perform an oral health assessment and oral care procedures. Prior to this, all participants received the two assessment sections (Parts I and II) for preparation. They were then scheduled to conduct the assessments and procedures in the laboratory at the tertiary healthcare training institution where the study was conducted. Participants were paired and instructed to perform oral health assessments and oral care procedures on each other. On the day of the practice, they were asked to sign an informed consent form. Two assessors independently assessed and rated all oral care procedures to ensure the appropriateness and applicability of each item. The oral care procedure was rated using three categories—‘unsatisfactory’, ‘satisfactory’, and ‘not done’—based on the subject’s performance. 

After completing the procedures, all participants were asked to complete the oral health assessment section (Part I). They were also asked if they encountered difficulties filling out the two sections. Four subjects raised a concern regarding the oral condition of older residents, as while it could be expected, there was no option to indicate this in Part I. To address this, an additional choice indicating a ‘normal’ condition was added to each item in Part I. A footnote was added to the denture assessment, stating ‘Older residents reported daily use of dentures to eat and speak, and denture(s) fits reasonably well in the mouth and the denture(s) does(do) not appear to be faulty/defective,’ to provide clear instructions on the appropriate use and function of dentures. Moreover, the third reviewer (LWK) recommended adding a description for Item 9 of Part II to clearly instruct the healthcare workers to implement this item, ‘Removed denture(s) during oral hygiene assessment shall be cleaned by a soft-bristle toothbrush with mild soap solution, thoroughly rinsed before re-inserted into mouth afterward’. Additionally, the two assessors recommended adding two more items to Part II to enhance the appropriateness and practicability of the tool: ‘Maintain communication with the older resident during the procedure’ and ‘Provide brief education and/or oral condition information to the older resident after the procedure’. The assessment tool was finalised as shown in Appendix A.

## 4. Discussion 

Older residents often have poorer oral health, attributed to an inability to perform self-care and inadequate oral health assessment and care provided by healthcare workers [28]. This is concerning as poor oral health can lead to more severe health problems, including non-communicable diseases and nutritional deficiencies [6,9,10]. Therefore, healthcare workers must be appropriately trained, guided, and monitored in older residents’ oral health assessment and care procedures.

This study developed an assessment tool with two sections specifically for healthcare workers who provide oral care to older residents. The present study focused on analysing the content validity and applicability of the items in the oral health and care procedure assessment sections to ensure their appropriateness. The items in both sections were structured based on a comprehensive literature review and following advice from dental and nursing experts [39,40]. The structured assessment sections were finalised through multiple rephrasing, modification, and expert verification stages. Part I comprises the oral health assessment, comprehensively assessing a resident’s oral condition. Part II comprises the oral care procedure, providing a systematic and comprehensive procedure for healthcare workers. 

To ensure the accuracy and applicability of all items used in the oral health assessment and oral care procedure assessment, six experts from various dental and nursing fields reviewed commented on, rephrased, and modified the items. Their input ensured the tool’s accuracy and relevance and enhanced its applicability and feasibility for healthcare workers.

The content validity of the tool was assessed using I-CVI, S-CVI, and S-CVI/UA scores, which indicated that the items were relevant and clear [38,39]. This tool allows healthcare workers to appropriately and effectively assess the oral conditions of older residents. The tool recommends that any abnormalities found during the assessment be highlighted and described in detail, and healthcare workers must report them to their supervisor and seek dental advice after the procedure is completed.

All recruited subjects in the pilot study had received training in oral health assessment and care procedures, and some had practical experience in performing oral care procedures. Conducting the pilot study was crucial for enhancing the validity and applicability of the two assessment sections. Comments from the nurse working in the LTCI contributed to improving the content reliability and applicability of the tool. The pilot study played a vital role in confirming the accuracy of the items and the assessment tool’s applicability and feasibility [39,41]. In addition, having two independent assessors evaluate the oral care procedures further enhanced the accuracy and applicability of the tool for oral health and oral care procedures assessment. The completion of the assessment tool typically takes approximately 10 min.

Oral health assessments are often overlooked; therefore, Part I for the oral health assessment will go with Part II for the oral care procedure to remind healthcare workers to complete each step. This assessment is a valuable guide for healthcare workers, providing step-by-step instructions for oral care for older residents. Additionally, it serves as a reminder to follow necessary precautions (for example, Item 2: ‘Use standard precautions and appropriate infection control measures during the oral care procedure’).

The finalised tool has been developed through a collaborative process involving expert consensus and a pilot study and will give healthcare workers more confidence in providing oral care for older residents. This tool can be utilised as a self-evaluation checklist or as an evaluation checklist for regular assessments used for training purposes regarding the oral care procedures performed by healthcare workers. It is crucial to address any unsatisfactory performance on specific items as they can directly impact the oral health of older residents. Healthcare workers who consistently receive unsatisfactory ratings should attend an oral health programme to enhance their knowledge and skills. 

The oral health assessment section and oral care procedure assessment section are interconnected and should be performed together. During the oral care procedures, the oral health assessment should be conducted to identify any dental consultation needs, and an oral evaluation should be carried out to determine if they require assistance in their daily oral care routine [29].

This assessment tool is not limited to LTCIs and can be used in non-dental clinical and community settings. Healthcare providers, including healthcare workers, nurse assistants, and nurses, should receive training on effective and appropriate oral health assessments and oral care procedures [29] as these are crucial for preventing complications associated with poor oral health in clients who cannot perform these procedures alone. 

This study highlights the significance of having an assessment tool for healthcare workers’ oral health and care procedures. There has been increased emphasis on oral health and dental care, particularly among older populations. The Outreach Dental Care Programme was launched in Hong Kong in 2014, offering free oral health services, dental check-ups, scaling, pain relief, fillings, tooth extractions, and dentures to older residents in LTCIs [6,43]. Therefore, this validated assessment tool is crucial for helping identify the oral problems experienced by older residents, and it should be used regularly. Additionally, the government should allocate additional resources for oral health and oral care skill training for healthcare workers to enhance their competence in oral health practice. LTCI management teams should also ensure that healthcare workers perform adequate and proper oral health assessments and oral care procedures. This assessment tool for oral health and oral care procedures should become a standard part of daily oral care practice to maintain the oral health of older residents in LTCIs.

### 4.1. Future Directions for Research and Tool Refinement 

This study has developed a validated assessment tool for healthcare workers to assess oral health status and conduct oral care procedures. However, there are several potential future directions for research and tool refinement to enhance the assessment of oral health status and oral care procedures. First, additional validation is needed for tools designed to evaluate denture-related issues and assess the risk of oral health complications. Second, customisation of the assessment tool based on the level of care dependence of older residents is essential to meet different needs and challenges. By considering variations in oral care practices across different settings, the assessment tools can be modified to meet the specific requirements of healthcare workers in each context. In addition, continuous refinement of the assessment tool for oral health and oral care procedures should be pursued through comparative analysis studies. By comparing the strengths and limitations of existing tools, researchers can guide the development of more comprehensive and accurate assessment methods. Moreover, longitudinal studies with a larger sample of healthcare workers are crucial to assess the long-term impact of oral care practices on the oral health outcomes of older residents to evaluate the effectiveness of current practices and refine assessment tools to result in long-term oral health improvements. 

Another potential direction is exploring the integration of technology, such as mobile applications or wearable devices. By incorporating features such as real-time data collection, automated analysis, and personalised feedback, the accuracy and efficiency of oral health and oral care procedures assessments can be enhanced. Collaborating with professionals from other disciplines, such as dental, geriatrics, or nursing, is critical to contribute to a comprehensive evaluation of oral health and care practices. Efforts towards standardising assessment tools and protocols across healthcare institutions and settings should be considered to enable greater data comparability, facilitate research collaboration, and promote consistent and evidence-based oral care practices.

### 4.2. Strengths and Weaknesses

This study has successfully developed and validated an oral health assessment tool specifically designed to evaluate the oral health condition of older residents and assess the oral care procedures performed by healthcare workers. This addresses an existing gap in oral health assessments for older residents as previous assessment tools have not adequately considered this population’s unique needs and challenges. This tool provides a systematic approach to help healthcare workers perform oral care practices more appropriately in their daily practice, ultimately improving the oral health outcomes of older residents. Additionally, this tool can benefit other healthcare professionals responsible for oral care and can be used to train caregivers providing home-based care for older individuals. It offers a standardised and comprehensive approach to oral health assessment and training, improving the competence of healthcare workers in oral care practice and leading to better oral health outcomes for older residents.

However, it is important to note that the study could not include the targeted group of older residents due to practical and ethical considerations. To enhance the validity and applicability of the assessment tool, it is recommended to conduct future studies that include these residents as participants. This would allow for a more comprehensive validation process and ensure that the tool accurately captures this specific population’s unique oral care needs and experiences. 

While this tool is specifically for older populations, modifications and revalidation may be necessary before using the tool in other populations. Additionally, further factor analysis is deemed infeasible due to the small sample size included in this study. To ensure the robustness and reliability of the tool, a larger sample size of healthcare workers is required for future studies.

## 5. Conclusions

This tool represents a significant advancement in oral health assessment and care for older populations. It is the first tool to incorporate assessments of oral health and of oral care procedures, and its development involved a comprehensive process, including an extensive literature review, expert review, multiple modification stages, tool refinement, and a pilot study to verify and validate its accuracy and applicability. The tool demonstrated high content validity and applicability and is not limited to older residents in LTCIs but can be used with other older populations in clinical and community settings. Its versatility makes it valuable for research and practice, with the potential to improve the oral health of older populations. Furthermore, it can serve as an evaluation instrument for regular staff training and assessing healthcare worker performance. The oral health assessment form can guide healthcare workers in conducting comprehensive oral assessments for dental management, while the oral care procedure checklist facilitates the evaluation of the procedures performed by healthcare workers. By using this tool, the quality of oral care provided to older residents can be enhanced, leading to improved overall oral health and well-being. To ensure the successful implementation of this tool, it is essential for the government to actively participate and provide adequate resources for oral health and oral care assessment training. This investment will improve oral health outcomes for older individuals, ultimately leading to a better quality of life for this population.

## Figures and Tables

**Figure 1 healthcare-12-00558-f001:**
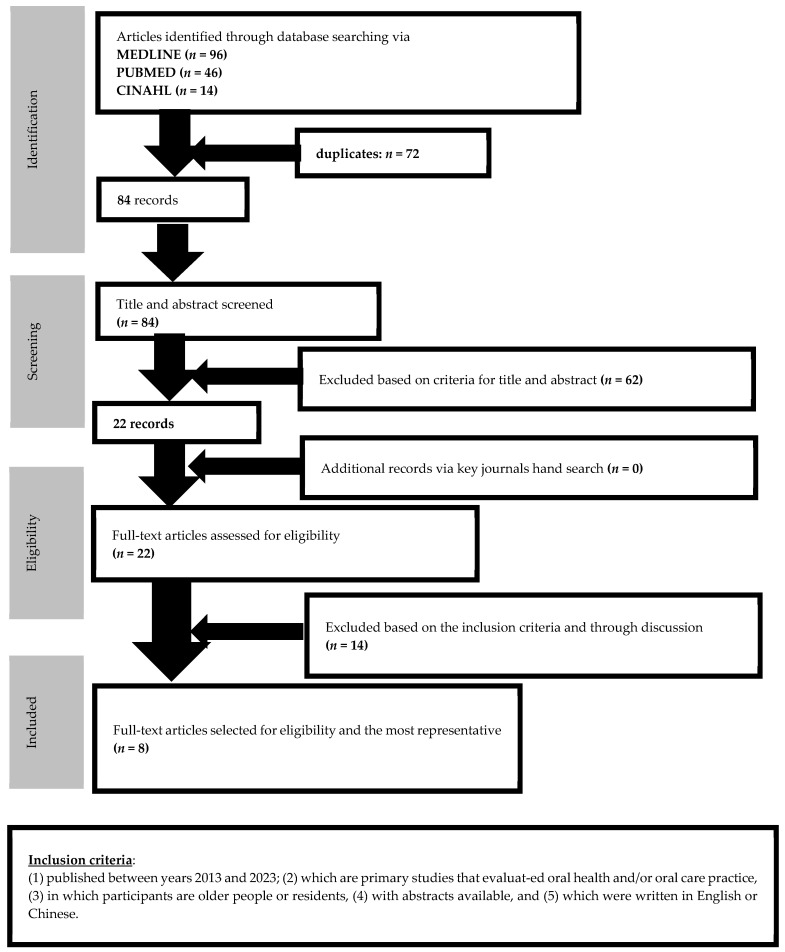
Flow of searching and Inclusion of relevant articles.

## Data Availability

The data presented in this study are available on request from the corresponding author. The data are not publicly available due to participants’ confidentiality.

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
