# Peer review of "Validating an Assessment Tool for Oral Health and Oral Care Procedures Performed by Healthcare Workers for Older Residents in Long-Term Care Institutions"

_healthcare, 2024, doi:10.3390/healthcare12050558_

Round 1

Reviewer 1 Report

Comments and Suggestions for Authors

The introduction provides a comprehensive overview of the global aging population, highlighting the increasing concern and projections for older individuals. It effectively sets the stage for the specific focus on oral health, emphasizing its close relationship with overall health.

The methods section is well-organized, detailed, and follows best practices in research methodology. The inclusion of expert consultations, rigorous validation procedures, and a pilot study contributes to the robustness of the assessment tool development process.

The section discussing strengths and weaknesses is beneficial for readers. However, it would be helpful to expand on the strengths further. For example, emphasize how the tool addresses an existing gap in oral health assessment for older residents.

The conclusion is well-structured and emphasizes the significance of the developed tool. To enhance it, consider summarizing the key findings and contributions made by the study. Additionally, restate the practical implications and potential impact of the tool on improving oral health for older individuals.

Consider adding a brief section on potential future directions for research and tool refinement. This could include suggestions for additional validation studies, modifications based on feedback, or adaptations for specific healthcare settings.

Overall, the discussion is informative, and expanding on the strengths, limitations, and practical implications will further enhance its depth and clarity.

Author Response

Thank you for your valuable comments. Please kindly find our replies below.

The introduction provides a comprehensive overview of the global aging population, highlighting the increasing concern and projections for older individuals. It effectively sets the stage for the specific focus on oral health, emphasizing its close relationship with overall health.

Reply: Thank you for your comment.

The methods section is well-organized, detailed, and follows best practices in research methodology. The inclusion of expert consultations, rigorous validation procedures, and a pilot study contributes to the robustness of the assessment tool development process.

Reply: Thank you for your comment.

The section discussing strengths and weaknesses is beneficial for readers. However, it would be helpful to expand on the strengths further. For example, emphasize how the tool addresses an existing gap in oral health assessment for older residents.

Reply: Thank you for your valuable comment. The strengths of the assessment tool for evaluation in daily practice and training are added in the section “Strengths and Weaknesses”.

The conclusion is well-structured and emphasizes the significance of the developed tool. To enhance it, consider summarizing the key findings and contributions made by the study. Additionally, restate the practical implications and potential impact of the tool on improving oral health for older individuals.

Reply: The key findings and contributors of the tool are highlighted in the “conclusion.” It’s practical implications and potential impact on improving oral health for older residents are emphasized.

Consider adding a brief section on potential future directions for research and tool refinement. This could include suggestions for additional validation studies, modifications based on feedback, or adaptations for specific healthcare settings.

Reply: Thank you for your suggestion. A brief section on potential future directions for research and tool refinement has been added before “strengths and weaknesses”

Overall, the discussion is informative, and expanding on the strengths, limitations, and practical implications will further enhance its depth and clarity.

Reply: Thank you.

Reviewer 2 Report

Comments and Suggestions for Authors

In your abstract, what are I-CVI, S-CVI, and S-CVI/UA scores?? describe

The introduction can be minimized

The methodology is unclear on how dental examination was done and how was the calibration with the trainers .... this part needs to be emphasized

I suggest you revise the English language as well

Comments on the Quality of English Language

I suggest you revise the English language with a native speaker or agency

Author Response

Thank you for your comments. Please kindly find our replies below.

In your abstract, what are I-CVI, S-CVI, and S-CVI/UA scores?? Describe

Reply: The I-CVI, S-CVI and S-CVI.UA scores were 1, indicating that excellent agreement among the experts on the relevance and clarity of all items in the tool. Therefore, the developed assessment tool for oral health and oral care procedure was validated with excellent content validity. The abstract was revised.

The introduction can be minimized

Reply: The introduction was shortened.

The methodology is unclear on how dental examination was done and how was the calibration with the trainers .... this part needs to be emphasized

Reply: Thank you for your comment. In this study, both oral health assessment and oral care procedure were conducted during the pilot study. The section “Pilot study” was revised with clearer methodology on ow the pilot study was conducted. The participants were the enrolled nurses or undergraduate students who has previously received training on oral care practices as their part of studies. Since the aim of this study was to validate the developed assessment tool for healthcare providers, there was no dedicated trainer in the study. However, the assessors involved in the procedure were experienced registered nurses and nurse educators with one being an Associate Professor and the other a senior clinical educator.

I suggest you revise the English language as well

Reply: The revised manuscript has been edited to improve English proficiency.

Reviewer 3 Report

Comments and Suggestions for Authors

Validation of Assessment Tool for Oral Health and Oral Care Procedure Performed by Healthcare Workers for Older Residents in Long-Term Care Institutions

L39 better to say with negative impact on QoL rather than poorly affecting.

L 42 had complete tooth loss or edentulous is better written as were totally edentulous.

2.6. A Pilot Study & 3.4. A Pilot Study

L 205-206 For participant recruitment, approximately 15-30 nursing students who meet the following criteria were selected: 1) aged 18 or older, 2) either nurse students who had ..

L 347 The mean age was 24.2 years old (SD 3.74) Participants were paired up and given instructions to perform oral health assessments and oral care procedures on each other.

The target for this Tool are the elderly in LTCIs. Yet the procedures were undertaken on young nurses who did not represent the intended group. Why were older people not involved as this would have been much more realistic & would flag areas for improvement in the Tool?

Under Part I. Oral Health Assessment. 5. Tooth should be Teeth. Normal & Count >20. 6. Denture(s) Normal. What does Normal mean? Other non-dental professionals may be confused with these terms.

Under Part II. Assessment of Oral Care Practice, there is no denture care section other than (9) Remove dentures if present. Given that many older patients will be wearing removable dentures, some form of denture care/hygiene should have been included in the procedure list.

L 302-304 The six experts reviewed and rated all items in the preliminary version of the content validity index (CVI). Their scores ranged from 4 to 5, and modifications were made based on their narrative comments.

This is not easily understood as CVR precedes CVI. Each domain will have a CVR & the CVI calculated by adding each CVR dividing by the total number of domains.

L 435 This assessment is a valuable guide for healthcare workers, providing step-by-step instructions for oral care for older residents. Step (10) Perform oral care thoroughly, including tongue, teeth, gum, oral membrane, and lips does NOT state how this is done. This implies the soft tissues may be cleaned in the same way as the teeth. A separate box for teeth should have been developed.

Table S2 is the most relevant as it shows both CVR & CVI. It must form part of the main paper. Why is CVI shown before CVR. Each Ratio should be shown first. It is a perfect 1.00 for every domain! Perhaps the authors should have included the initial CVRs & CVIs as well as the final ones. Readers will then see which domains were improved.

Why do the authors think this Tool is only applicable to the elderly? For instance, there is no section regarding physical disability or inability to self-perform oral procedures. How does it improve on OHAT? There is no discussion on the advantages of this Tool compared to others.

The authors have introduced & developed another Oral Assessment Tool without providing a rationale for this other than its use for the elderly. But the Tool domains are not specific to the older patient. Furthermore, OHAT gives a score for each domain and an overall score which helps indicate treatment need. This new Tool does not.

References incorrectly spelt names

L 564 19 Stančić, I.; Petrovi´c, M.; Popovac, A.; Vasovi´c, M.; Despotovi´c, N. Caregivers’ attitudes, knowledge and practices of oral care at nursing homes in Serbia. Vojnosanit. Pregl. 2016, 73, 668–673. doi: 10.2298/VSP141001065S.

Should be Petrović………. Vasović…………..Despotović. The accent above ć can be omitted.

Validation of Assessment Tool for Oral Health and Oral Care Procedure Performed by Healthcare Workers for Older Residents in Long-Term Care Institutions

L39 better to say with negative impact on QoL rather than poorly affecting.

L 42 had complete tooth loss or edentulous is better written as were totally edentulous.

2.6. A Pilot Study & 3.4. A Pilot Study

L 205-206 For participant recruitment, approximately 15-30 nursing students who meet the following criteria were selected: 1) aged 18 or older, 2) either nurse students who had ..

L 347 The mean age was 24.2 years old (SD 3.74) Participants were paired up and given instructions to perform oral health assessments and oral care procedures on each other.

The target for this Tool are the elderly in LTCIs. Yet the procedures were undertaken on young nurses who did not represent the intended group. Why were older people not involved as this would have been much more realistic & would flag areas for improvement in the Tool?

Under Part I. Oral Health Assessment. 5. Tooth should be Teeth. Normal & Count >20. 6. Denture(s) Normal. What does Normal mean? Other non-dental professionals may be confused with these terms.

Under Part II. Assessment of Oral Care Practice, there is no denture care section other than (9) Remove dentures if present. Given that many older patients will be wearing removable dentures, some form of denture care/hygiene should have been included in the procedure list.

L 302-304 The six experts reviewed and rated all items in the preliminary version of the content validity index (CVI). Their scores ranged from 4 to 5, and modifications were made based on their narrative comments.

This is not easily understood as CVR precedes CVI. Each domain will have a CVR & the CVI calculated by adding each CVR dividing by the total number of domains.

L 435 This assessment is a valuable guide for healthcare workers, providing step-by-step instructions for oral care for older residents. Step (10) Perform oral care thoroughly, including tongue, teeth, gum, oral membrane, and lips does NOT state how this is done. This implies the soft tissues may be cleaned in the same way as the teeth. A separate box for teeth should have been developed.

Table S2 is the most relevant as it shows both CVR & CVI. It must form part of the main paper. Why is CVI shown before CVR. Each Ratio should be shown first. It is a perfect 1.00 for every domain! Perhaps the authors should have included the initial CVRs & CVIs as well as the final ones. Readers will then see which domains were improved.

Why do the authors think this Tool is only applicable to the elderly? For instance, there is no section regarding physical disability or inability to self-perform oral procedures. How does it improve on OHAT? There is no discussion on the advantages of this Tool compared to others.

The authors have introduced & developed another Oral Assessment Tool without providing a rationale for this other than its use for the elderly. But the Tool domains are not specific to the older patient. Furthermore, OHAT gives a score for each domain and an overall score which helps indicate treatment need. This new Tool does not.

References incorrectly spelt names

L 564 19 Stančić, I.; Petrovi´c, M.; Popovac, A.; Vasovi´c, M.; Despotovi´c, N. Caregivers’ attitudes, knowledge and practices of oral care at nursing homes in Serbia. Vojnosanit. Pregl. 2016, 73, 668–673. doi: 10.2298/VSP141001065S.

Should be Petrović………. Vasović…………..Despotović. The accent above ć can be omitted.

Comments on the Quality of English Language

Author Response

Reviewer 3

Thank you for your valuable comments. Please kindly find our replies below.

L39 better to say with negative impact on QoL rather than poorly affecting.

Reply: The sentence has been revised according to the comment.

L 42 had complete tooth loss or edentulous is better written as were totally edentulous.

Reply: The sentence has been revised according to the comment.

2.6. A Pilot Study & 3.4. A Pilot Study

L 205-206 For participant recruitment, approximately 15-30 nursing students who meet the following criteria were selected: 1) aged 18 or older, 2) either nurse students who had ..

L 347 The mean age was 24.2 years old (SD 3.74) Participants were paired up and given instructions to perform oral health assessments and oral care procedures on each other.

The target for this Tool are the elderly in LTCIs. Yet the procedures were undertaken on young nurses who did not represent the intended group. Why were older people not involved as this would have been much more realistic & would flag areas for improvement in the Tool?

Reply: Thank you for your comment and reminder. This study focused on validating a tool designed to assess oral care practices for older residents, with healthcare workers as the participants. However, due to practical and ethical considerations, the targeted group of older residents could not be included in the study. This limitation is important to acknowledge, as the validation process should ideally involve the intended population.

To enhance the validity and applicability of the assessment tool, it is recommended to conduct future studies that include older residents as participants. This would allow for a more comprehensive validation process and ensure that the tool accurately captures the unique oral care needs and experiences of this specific population.

This limitation has been added to the "Strengths and Weaknesses" section, emphasizing the importance of validating the assessment tool with older residents in future research to further enhance its validity and applicability to the target population.

Under Part I. Oral Health Assessment. 5. Tooth should be Teeth. Normal & Count >20. 6. Denture(s) Normal. What does Normal mean? Other non-dental professionals may be confused with these terms.

Reply: Thank you for your kind reminder. For the ‘denture(s), normal means without abnormal listed as abnormal items. To clarify the meaning of ‘Normal”, a footnote “older residents reported daily use of dentures to eat and speak; and denture(s) fits reasonably well in the mouth and does(do) not appear to be faulty/defective.” has been added.

The design of this assessment tool was simple as possible. The tool should be used after education is given.

Under Part II. Assessment of Oral Care Practice, there is no denture care section other than (9) Remove dentures if present. Given that many older patients will be wearing removable dentures, some form of denture care/hygiene should have been included in the procedure list.

Reply: Thank you for your comment. #9 item has been enriched according to the comment. Further explanation has been added, “Removed denture(s) during oral hygiene assessment shall be cleaned by a soft bristle toothbrush with mild soap solution, thoroughly rinsed before reinserted into mouth afterwards,”  in the item to increase the understanding.

L 302-304 The six experts reviewed and rated all items in the preliminary version of the content validity index (CVI). Their scores ranged from 4 to 5, and modifications were made based on their

This is not easily understood as CVR precedes CVI. Each domain will have a CVR & the CVI calculated by adding each CVR dividing by the total number of domains.

Reply:  Thank you for your comment. It is not correct because a panel of experts wh rate each item for its relevance to the construct being measured to calculate the content validity ratio (CVR). The CVR is calculated by determining the number of experts who rate the item as essential or relevant divided by the total number of experts. This paragraph has been revised more clearly.

L 435 This assessment is a valuable guide for healthcare workers, providing step-by-step instructions for oral care for older residents. Step (10) Perform oral care thoroughly, including tongue, teeth, gum, oral membrane, and lips does NOT state how this is done. This implies the soft tissues may be cleaned in the same way as the teeth. A separate box for teeth should have been developed.

Reply: Thank you for your comment. The #10 was modified. To ensure thorough oral care, the item for teeth was not separated as another item. This approach is intended to ensure that all aspects of oral care are covered comprehensively and to avoid overlooking any particular area. The item is designed to encompass all aspects of teeth care, including brushing and flossing, as well as wiping the soft tissue with a moist gauze.

Table S2 is the most relevant as it shows both CVR & CVI. It must form part of the main paper. Why is CVI shown before CVR. Each Ratio should be shown first. It is a perfect 1.00 for every domain! Perhaps the authors should have included the initial CVRs & CVIs as well as the final ones. Readers will then see which domains were improved.

Reply: Thank you for your comment. The table S2 has been revised with the CVR shown before CVI. Table S2 is the initial CVRs and CVIs. The recommendations were made by the experts for improvement. The final table is showed in Table 1.

Why do the authors think this Tool is only applicable to the elderly? For instance, there is no section regarding physical disability or inability to self-perform oral procedures. How does it improve on OHAT? There is no discussion on the advantages of this Tool compared to others.

Reply: This assessment tool was developed for healthcare workers instead of older residents at LTCIs. Therefore, the details of ability of self-perform oral practices were not included. Because this tool is for healthcare workers, the advantages were described in the “discussion”.

The authors have introduced & developed another Oral Assessment Tool without providing a rationale for this other than its use for the elderly. But the Tool domains are not specific to the older patient. Furthermore, OHAT gives a score for each domain and an overall score which helps indicate treatment need. This new Tool does not.

Reply: The reason for developing this assessment tool was explained in the "Introduction" section. In order to clearly state the purpose, the last two paragraphs of the "Introduction" have been revised. The developed assessment tool includes screening items that are specifically relevant for older individuals, such as assessing denture condition. Unlike the OHAT, the developed assessment tool does not include scoring because all abnormalities identified should prompt a referral for dental consultation.

References incorrectly spelt names

L 564 19 Stančić, I.; Petrovi´c, M.; Popovac, A.; Vasovi´c, M.; Despotovi´c, N. Caregivers’ attitudes, knowledge and practices of oral care at nursing homes in Serbia. Vojnosanit. Pregl. 2016, 73, 668–673. doi: 10.2298/VSP141001065S.

Should be Petrović………. Vasović…………..Despotović. The accent above ć can be omitted.

Reply: Thank you. The names have been revised.

Round 2

Reviewer 2 Report

Comments and Suggestions for Authors

The introduction focuses on HK, it is advised to include other global comparison since you intend to publish in an international journal.

In your results, what do you mean by SD?

Please minimize your limitations to things that are specific and not general. As for your recommendations, think of other healthcare providers as study participants.

Based on your pilot study, you should recommend inclusion of more participants .. In addition, think of involvement of elders.

Comments on the Quality of English Language

Better

Author Response

Thank you for your comments again. Please kindly find our replies below.

The introduction focuses on HK, it is advised to include other global comparison since you intend to publish in an international journal.

Reply: the content related to the oral practice performed by healthcare workers has been added to compare the practice in Hong Kong. Please find the additional content highlighted in green in the Introduction section.

In your results, what do you mean by SD?

Reply:  The full term of SD, ‘standard deviation’ has been added. Please find the revision highlighted in green on page 9.

Please minimize your limitations to things that are specific and not general. As for your recommendations, think of other healthcare providers as study participants.

Reply:  The ‘Strengths and Limitations’ has been minimized to be more specific accordingly. Please find the revision highlighted in green in this section.

            Regarding the recommendation of the recruitment of other healthcare providers as study participants has been added in section 4.1 ‘Future Directions for Research and Tool Refinement’ and the revision has been highlighted in green.

Based on your pilot study, you should recommend inclusion of more participants .. In addition, think of involvement of elders.

Reply:   An increase in sample size and the involvement of older individuals have been added in the recommendation in future studies according to your comment.

Reviewer 3 Report

Comments and Suggestions for Authors

The script is acceptable now.

Author Response

Thank you for accepting our revised manuscript.